# National Early Warning Scores and COVID-19 deaths in care homes: an ecological time-series study

Daniel Stow [ID], Robert O Barker, Fiona E Matthews, Barbara Hanratty [ID]

Population and Health Sciences Institute, Newcastle University, Newcastle upon Tyne, UK

**Correspondence to**
Daniel Stow;
daniel.stow@ncl.ac.uk

## ABSTRACT

**Objectives** To investigate whether National Early Warning Scores (NEWS/NEWS2) could contribute to COVID-19 surveillance in care homes.

**Setting** 460 care home units using the same software package to collect data on residents, from 46 local authority areas in England.

**Participants** 6464 care home residents with at least one NEWS recording.

**Exposure measure** 29 656 anonymised person-level NEWS from 29 December 2019 to 20 May 2020 with component physiological measures: systolic blood pressure, respiratory rate, pulse rate, temperature and oxygen saturation. Baseline values for each measure calculated using 80th and 20th centile scores before March 2020.

**Outcome measure** Cross-correlation comparison of time series with Office for National Statistics weekly reported registered deaths of care home residents where COVID-19 was the underlying cause of death, and all other deaths (excluding COVID-19) up to 10 May 2020.

**Results** Deaths due to COVID-19 were registered from 23 March 2020 in the local authority areas represented in the study. Between 23 March 2020 and 10 May 2020, there were 5753 deaths (1532 involving COVID-19 and 4221 other causes). We observed a rise in the proportion of above-baseline NEWS beginning 16 March 2020, followed 2 weeks later by an increase in registered deaths (cross-correlation of $r=0.82$, $p<0.05$ for a 2 week lag) in corresponding local authorities. The proportion of above-baseline oxygen saturation, respiratory rate and temperature measurements also increased approximately 2 weeks before peaks in deaths.

**Conclusions** NEWS could contribute to COVID-19 disease surveillance in care homes during the pandemic. Oxygen saturation, respiratory rate and temperature could be prioritised as they appear to signal rise in mortality almost as well as NEWS. This study reinforces the need to collate data from care homes, to monitor and protect residents' health. Further work using individual level outcome data is needed to evaluate the role of NEWS in the early detection of resident illness.

## INTRODUCTION

Care homes have experienced high rates of COVID-19 infection and death. In England, over half of excess mortality in the first months of 2020 is estimated to have been

## Strengths and limitations of this study

► This is one of the few studies providing population level surveillance information on the care home population during the initial peak of infections from COVID-19.

► This study uses widely available public health mortality information, combined with individual level health observations based on vital sign measurements (National Early Warning Scores), that could be useful for population-level COVID-19 disease surveillance during future waves of COVID-19 infection in care homes.

► The ecological study design was used as a pragmatic approach to make best use of available data: but this is not a causal study, nor a study of diagnostic accuracy, and it is liable to the ecological fallacy.

► Further research using individual level information on mortality and diagnoses, linked to NEWS, is required to evaluate the role of NEWS in assessing individual residents with suspected COVID-19 infection.

in this setting.[1] Surveillance of COVID-19 in care homes has been difficult, because of a paucity of testing, and the lack of experience with how this disease presents in older people.[2]

An increasing number of UK care homes now collect National Early Warning Score (NEWS). This tool (originally developed by the Royal College of Physicians in 2012,[3] and updated in 2017)[4] is used in hospitals in the UK to identify patients at risk of acute deterioration and improve patient safety. NEWS requires the measurement of six simple physiological parameters: temperature, pulse, systolic blood pressure, respiratory rate and oxygen saturation and level of consciousness or new confusion. The score generated should trigger a prespecified response, ranging from repeating the score within a specific timeframe, to seeking urgent medical attention.[5]

The British Geriatrics Society produced guidelines for managing COVID-19 in care

homes for older people. To support triage, they recommend training staff to measure a resident's vital signs (temperature, blood pressure, heart rate, pulse oximetry and respiratory rate) when COVID-19 infection is suspected.[6] The risks and benefits of this approach are unknown. In community settings, elevated NEWS scores have been associated with prompt review by a health professional and poor health outcomes.[7] Evidence to support the use of NEWS in care homes is limited, and whether it can support care home staff to identify or predict deterioration in the health of residents is unclear.[8–11] NEWS observations, particularly measurement of blood pressure, require close contact between residents and care home workers, which may transmit COVID-19 infection.

This study will address the dearth of published evidence on the use of NEWS in any setting during the COVID-19 pandemic.[12] The aim is to describe change in NEWS and its components over time and align these data with COVID-19 and all-cause mortality in care homes in England. We address the question of whether NEWS can contribute to surveillance during the pandemic, and whether an abbreviated NEWS (excluding one or more of the component measures) would suffice.

## METHODS
### Study design and setting
We conducted an ecological study by aggregating individual level data for the exposure of interest (NEWS) and comparing these results to area level aggregate data for the outcome of interest (all cause and COVID-19 mortality). Participants were all residents of care homes utilising the same commercial software with cloud storage of data, and they all had at least one NEWS recording.[13] We did not apply any further population inclusion/exclusion criteria.

### Exposure information
Anonymised person-level NEWS (and the slightly modified NEWS2)[4] information were obtained from 1 December 2019 to 20 May 2020 including the individual component vital sign measures of NEWS: blood pressure, respiratory rate, pulse rate, temperature and oxygen saturation. Demographic information on the care home resident (age and sex) were obtained, along with a geographical identifier to examine regional variation. NEWS recordings were made by care home staff using a specific, commercially available package designed to reduce measurement error. The commercial provider supplies equipment to measure vital signs and record physiological observations, automatically generate NEWS, and upload the data to their cloud storage servers. The system has previously been described elsewhere.[11]

### Outcome measurement
Geographical death data were obtained from Office for National Statistics (ONS) weekly reported registered deaths in care homes due to COVID-19, and all cause deaths (excluding COVID-19) available from week 1 (beginning 29 December 2019), to week 19 (ending 10 May 2020).[14] ONS reporting areas and care home geographical labels were mapped as closely as possible.

### Analysis
We established baseline levels for NEWS and its component observations, in our population using centile cut points in a random subset of 70% of data collected between December 2019 and 1 March 2020 (prior to the likely outbreak of COVID-19 in care homes in England). These cut points were then validated in the remaining 30% of observations made before March 2020. NEWS scores above the 80th centile score were defined as above-baseline. For individual physiological observations, upper and lower thresholds were established using the 80th and 20th centile measurements in the same random subset (we calculated the lowest 20th centile only for oxygen saturation: only lower values indicate a health problem). These cut points represent markers of increase in each parameter at a population level and are not measures of clinical concern at the individual resident level, which are defined elsewhere.[4] We removed biologically implausible values from the data set before creating the centile cut points (online supplemental table 1). We used quantile regression to measure the impact of age and sex on centile scores but did not find evidence to support age/sex specific 20th and 80th centile cut points in this population (details available from the authors on request).

We calculated the proportion of above-baseline NEWS measurements and the component observations on a weekly basis, aggregated across all geographical areas providing data. We plotted the proportion of weekly above-baseline measures in participating care homes as a time series against the weekly number of care home deaths due to COVID-19 and all-cause mortality (excluding COVID-19) occurring in the matched geographical areas present in our data.

We calculated the cross-correlation between the two time series (above-baseline NEWS and component scores, and daily registered all cause deaths including COVID-19) for time lags between 0 and 7 weeks. We combined the individual physiological observations that anticipated COVID-19 mortality trends to see if a 'minimum panel' could be useful for collection instead of the full NEWS panel. This makes our findings relevant to care home settings where NEWS is not calculated, or where all component physiological measures are not performed. Measurement of fewer components is expected to reduce contact time between carers and residents and decrease spread of COVID-19 infection.

All data management and analyses were conducted using R V.3.6.3.[15] We used the ccf function in the stats R package to calculate cross-correlations.

## Patient and public involvement

This was a study in response to a Public Health Emergency of International Concern. Patients or the public were not involved in the design, conduct or reporting of this rapid response research.

## RESULTS

### Care home population

Care home data were available from 6464 individuals, 2007 men (mean age 80.1 years, SD=12.6) and 3373 women (mean age 83.0 years, SD=12.9). Information on gender was missing from 1086 (16.8%) people, and age information was missing for 116 (1.8%) people. 441 biologically implausible NEWS component scores were removed from the dataset (online supplemental table 1).

### Geographical variation in reporting

29 656 NEWS recordings were made across 46 local authority (LA) areas, from 480 unique care home IDs (identifiers for the device used to record the measurement, representing a care home or a distinct unit within a care home). Most recordings were made in two LAs in the north east of England (n=11 029 and n=10 347), and in one London borough (n=3411).

### Deaths in care homes

There were 10 407 registered deaths in care homes in the 46 LA and Clinical Commissioning Group (CCG) areas between 29 December 2019 and 10 May 2020. The first death from COVID-19 was registered in week commencing 23 March 2020. From 23 March 2020 to 10 May 2020, there were 5753 deaths of care home residents—1532 with an underlying cause of COVID-19 and 4221 due to causes excluding COVID-19.

### Baseline news centile scores

Table 1 contains information on NEWS taken from 9586 (70% subset of 13 694) recordings made before 1 March 2020. Table 1 also contains thresholds for NEWS 20th centile scores, and 80th and 20th centile scores for NEWS components calculated in this subset, and the proportion of observations exceeding these values in the 30% validation dataset.

### NEWS measurements and care home deaths

The proportion of above baseline NEWS observations was stable from week 1 (30 December 2019–5 January 2020) until week 12 (16 March 2020 to 22 March 2020) and week 13 (23 March 2020 to 29 March 2020) when there was a marked increase. This increase happened in the weeks before the majority of COVID-19 and non-COVID-19 deaths began to occur (from week 15 (6 April to 12 April). The proportion of above baseline NEWS scores peaked in week 15, before beginning to decline again from week 16 (13 April to 19 April) onwards (figure 1). The highest correlation was observed for a 2 week lag (r=0.82, p≤0.05, online supplemental figure 1).

### Individual NEWS component measures and care home deaths

The proportion of above baseline measures of high respiratory rate (r=0.73, p≤0.05 for a 2 week lag) and low oxygen saturation (r=0.80, p≤0.05 for a 2 week lag) appear to follow the pattern of COVID-19 and non-COVID-19 deaths more closely than other component measures (online supplemental figure 1). The proportion of above baseline measures of temperature appeared to be decreasing between week 0 and week 10, before rising slightly to plateau until week 15 before declining again.

### Combination of NEWS component measures

Figure 2 shows paired combinations of above baseline respiratory rate and temperature and below baseline oxygen saturation. All increase just before peaks in COVID-19 and non COVID-19 deaths (figure 2).

## DISCUSSION

This study suggests that NEWS could make a useful contribution to COVID-19 disease surveillance in care homes during the pandemic. A rise in the proportion of above-baseline NEWS was observed from the middle of March 2020, when the incidence of COVID-19 was believed to be rising in the UK. The proportion of above-baseline

**Table 1** NEWS values before March 2020 in the development data set, cut points for baseline measurements and proportion of above baseline measurements in the validation data set

| Measurement | Development data set | | | | | | Validation data set | |
| | Median | Min | Max | IQR | 20th centile | 80th centile | % below 20th centile | % above 80th centile |
| --- | --- | --- | --- | --- | --- | --- | --- | --- |
| NEWS | 2.0 | 0.0 | 15.0 | 3.0 | | >4 | | 14.7 |
| Temperature (°C) | 36.5 | 32.0 | 40.0 | 0.6 | <36.2 | ≥36.9 | 18.4 | 25.8 |
| Pulse rate (beats/min) | 76.0 | 22.0 | 212.0 | 19.0 | <66 | ≥89 | 19.6 | 20.0 |
| Systolic BP (mm Hg) | 123.0 | 50.0 | 235.0 | 26.0 | <109 | ≥143 | 20.5 | 19.4 |
| Respiratory rate (breaths/min) | 19.0 | 6.0 | 60.0 | 3.0 | <17 | ≥22 | 19.4 | 18.3 |
| Oxygen saturation (%) | 95.0 | 47.0 | 100.0 | 4.0 | ≤92 | | 17.6 | |

BP, blood pressure; NEWS, National Early Warning Scores.

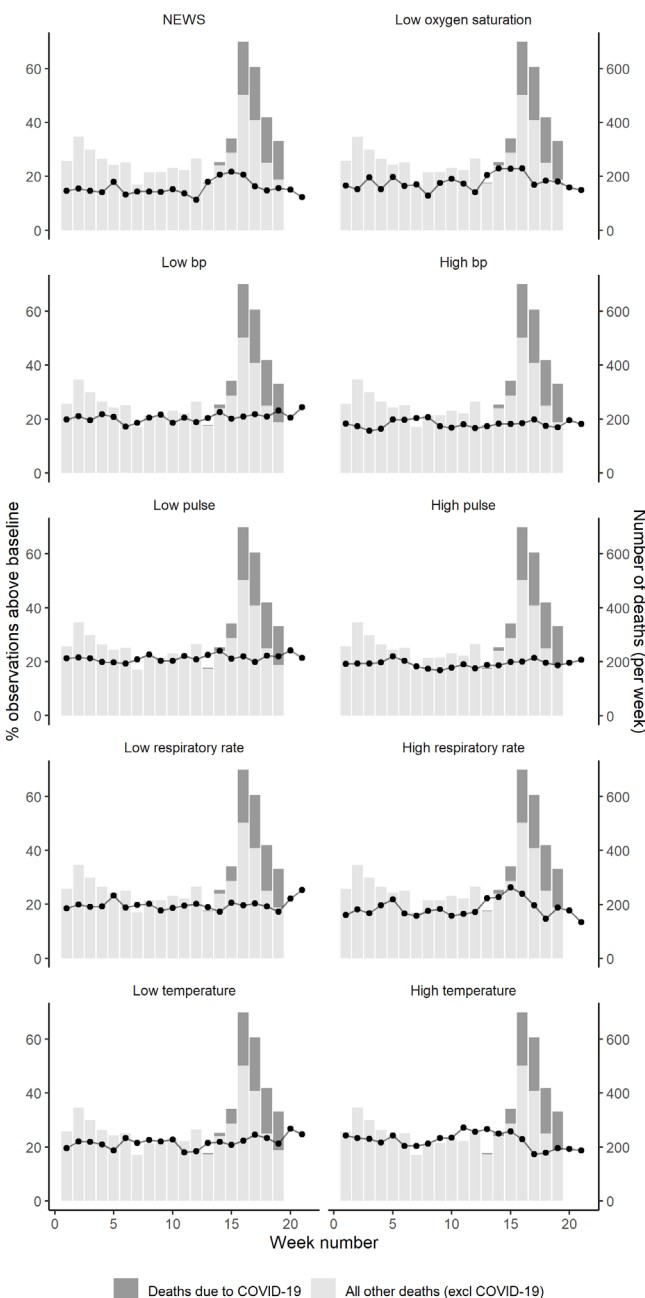

**Figure 1** The proportion of above baseline NEWS and component measurements from December 2019 to May 2020 (lines), compared with care home deaths in corresponding geographical areas in England (bars). NEWS, National Early Warning Scores.

measurements of oxygen saturation, respiratory rate and temperature also increased approximately 2 weeks before peaks in care home deaths in corresponding geographical areas. Oxygen saturation and respiratory rate appear to signal rise in mortality almost as well as total NEWS and may be safer and more practical to measure during a pandemic.

In this study, we observed a 2-week time lag between peaks in NEWS measures and deaths. This is similar to the observed time between symptom onset and

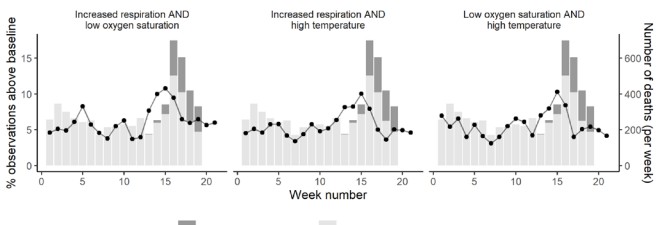

**Figure 2** The proportion of above baseline NEWS component combinations December 2019 to May 2020, compared with care home deaths in corresponding geographical areas in England. NEWS, National Early Warning Scores.

COVID-19 death in other settings.[16 17] Evidence for the role of NEWS in acute illness in community settings is growing,[11 18–22] but to date, only one descriptive study has provided empirical data on NEWS in care homes.[8] A recent systematic review on the use of NEWS in assessing unwell COVID-19 patients in primary care suggested that enthusiasm for its use may be premature.[12] Limitations of this study mean that we cannot draw direct conclusions about the role of NEWS (or its component parameters) in the care of individual residents: further study would be needed to address this question. However, it suggests a role for NEWS in COVID-19 disease surveillance at a care home population level in the pandemic. Measurement and monitoring of some NEWS components may be useful in detecting future waves of COVID-19 infection in care homes. Emerging evidence suggests that up to half of care home residents do not have symptoms at the time they test positive for COVID-19.[23] Whether NEWS measurement could signal impending illness in any of these residents at the individual level is unclear, and we do not believe it should substitute for a comprehensive programme of testing in care homes. Recent work has shown the value of routinely collected information to target resources during the COVID-19 pandemic,[24] and our study strengthens the case for further work to evaluate the role of NEWS in the care of individual residents.

### Strengths and limitations
To the best of our knowledge, this is the first study to examine variation in NEWS in care homes over time. We have described trends over time in NEWS recorded using a specific software system used to collect data on NEWS in some care homes. This means that the distribution of care homes within and between areas is not systematic, as it reflects the market share of the software company and local support for digital data collection in care homes. Most recordings were drawn from the north east of England, and a London borough, but we have no information on the proportion of care home residents in each area that are represented in our data set. All data were anonymised, and without individual outcome data, we examined patterns in and simple correlations between NEWS and area-level weekly registered death information. We were not able to independently assess or verify

the accuracy of NEWS measurements made by care home staff using the specific commercial software, and acknowledge that even where training is sufficient, the accuracy of vital sign measurement such as respiratory rate can be suboptimal.[25] However, our approach using 20th/80th centile scores will have ameliorated some of the impact of potential inaccuracies, and we caution against using single observations in isolation. Furthermore, we removed a small number of biologically implausible values and provide a summary of the number of values removed in online supplemental table 1. This study design was a pragmatic approach that made best use of available data, but it is not a causal study, nor a study of diagnostic accuracy and it is liable to the ecological fallacy.

## CONCLUSIONS

The recording of the NEWS, and the component physiological measures, may make a useful contribution to COVID-19 disease surveillance during the pandemic. Use of a shortened NEWS could be recommended for care home population surveillance where COVID-19 is of primary concern (but not for individual care due to the risk of underestimating the severity of non-respiratory illness). Oxygen saturation, respiratory rate and temperature provide a similar signal to the complete NEWS. The omission of some components of NEWS, such as blood pressure, minimises contact time between residents and care home staff, potentially reducing infection risk where this is of primary concern. Data on mortality and diagnoses, linked to NEWS, are required to evaluate the role of NEWS in assessing individual residents with suspected COVID-19 infection. Collection and aggregation of data from care homes would facilitate disease surveillance; we argue that introduction of a care home minimum dataset should be a priority for the UK.

**Acknowledgements** We are grateful to Solcom (Wyzan), who provided the anonymised NEWS information used in this study, the care home staff who collected the data, and the North East and North Cumbria Academic Health Science Network who support digital data collection in north east care homes.

**Contributors** BH and ROB conceived the study. BH, ROB and DS acquired and managed the data. DS analysed the data and created the figures. DS, ROB, FEM and BH interpreted the results. DS, BH and ROB drafted the first version of the manuscript. BH and FEM supervised the work. DS is the guarantor and accepts full responsibility for the work and the conduct of the study, had access to the data and controlled the decision to publish. All coauthors provided critical comments and approved the final version of the manuscript. The corresponding author attests that all listed authors meet authorship criteria and that no others meeting the criteria have been omitted.

**Funding** DS is funded by the NIHR School for Primary Care Research (Launching Fellowship SPCR-PDF-2020-161), ROB by an NIHR In Practice Fellowship (IPF-2018-12-010). BH is supported by the NIHR North East and North Cumbria Applied Research Collaboration. BH and FEM are funded by the NIHR Policy Research Unit: Older People and Frailty (PR-PRU-1217-21502). This paper presents independent research funded by the National Institute for Health Research. The views expressed are those of the author(s) and not necessarily those of the NHS, the NIHR or the Department of Health and Social Care.

**Competing interests** None declared.

**Patient consent for publication** Not required.

**Ethics approval** This study was approved by Newcastle University Research Ethics Committee (Ref. 3297/2020) and was conducted under the Secretary of State's directions under the Control of Patient Information Regulations, following advice from the Health Research Authority.

**Provenance and peer review** Not commissioned; externally peer reviewed.

**Data availability statement** Data are available in a public, open access repository. Data are available upon reasonable request. Information on the deaths of care home residents is feely available from the source cited (https://www.ons.gov.uk/peoplepopulationandcommunity/birthsdeathsandmarriages/deaths/articles/deathsinvolvingcovid19inthecaresectorenglandandwales/deathsoccurringupto1may2020andregisteredupto9may2020provisional). Other data may be available upon reasonable request from barbara.hanratty@newcastle.ac.uk.

**ORCID iDs**
Daniel Stow http://orcid.org/0000-0002-9534-4521
Barbara Hanratty http://orcid.org/0000-0002-3122-7190

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
