## [Reviewer comments · BMJ Open]

ARTICLE DETAILS

TITLE (PROVISIONAL)	National Early Warning Scores and COVID-19 deaths in care homes: an ecological time series study
AUTHORS	Stow, Daniel; Barker, Robert; Matthews, Fiona; Hanratty, Barbara

VERSION 1 – REVIEW

REVIEWER	Ram Gopal University of Warwick, Warwick Business School
REVIEW RETURNED	26-Oct-2020

GENERAL COMMENTS	I enjoyed reading this interesting paper. The work explores the use of NEWS as a mechanism for disease surveillance in care homes during the COVID-19 pandemic. A key finding is that an increase in the proportion of above-baseline NEWS scores is a 2-week advance indicator of COVID-19 cases. Also interesting is that the finding that just a subset of the measures, which do not require close contact with the patient to implement, suffices for surveillance purposes. The manuscript raises several questions which the authors must attempt to address: - It is difficult to understand the rationale for the omission of some components of NEWS. This data is collected anyway and what is the point of not using it for surveillance?- There does not seem to be a compelling reason to use 20th/80th centile scores. Why not simply follow the recommendations of the Royal College of Physicians? For example, they recommend “If the patient scores 3 in a component, regardless of the overall result, the patient is likely to require a higher level of care”.- When a patient has an above-baseline NEWS score it implies that the patient’s health has already deteriorated. And given that there is a 2-week gestation period for COVID-19, it follows that there will be a spike in cases two weeks hence. What additional insights are generated beyond this are generated in the analysis? The analysis may foretell the coming increase in cases but may not offer any preventive solutions as these patients are likely infected already. Would it not be more beneficial, from a preventive perspective, to forecast above-baseline NEWS scores so that appropriate actions can be taken to prevent the spread of the virus?
---

REVIEWER	Niam Yaraghi University of Miami, Business Technology
REVIEW RETURNED	05-Nov-2020

GENERAL COMMENTS	This is a very interesting and obviously timely paper in which the authors examine if there is any correlation between rise in NEWS measures and mortality in nursing homes in UK. I have the following suggestions to improve the paper:
---

	For the readers outside of the UK who are not familiar with the NEWS system, it would be very helpful if the authors include a section and explain the system, its history and maybe some areas that it has been proven effective before. Running this system smoothly requires significant effort in data collection and validation and I am very curious to know more about how the system is administrated and how the data is collected, verified and published. I was also expecting to see more robust time series analysis in which the association between the two series (NEWS measures and nursing home mortality) is statistically examined, but the authors just present the graph, with no statistical analysis of the association between the two. This is easily doable in common statistical software, for example, the authors can build timeseries models very conveniently using TSFS tool in SAS and examine how the two series are correlated and if such correlation is statistically significant. Of course there are more involved methods such as VARMAX which requires more efforts and probably additional data which authors can explore.
--	--

VERSION 1 – AUTHOR RESPONSE

REVIEWER 1	
I enjoyed reading this interesting paper. The works explores the use of NEWS as a mechanism for disease surveillance in care homes during the COVID-19 pandemic. A key finding is that an increase in the proportion of above-baseline NEWS scores is a 2-week advance indicator of COVID-19 cases. Also interesting is that the finding that just a subset of the measures, which do not require close contact with the patient to implement, suffices for surveillance purposes. The manuscript raises several questions which the authors must attempt to address:	Thank you for your succinct appraisal of the study – we have revised our manuscript with changes tracked, and copied relevant revisions in response to the queries you raise below
- It is difficult to understand the rationale for the omission of some components of NEWS. This data is collected anyway and what is the point of not using it for surveillance?	P8 – line 23: We have clarified that we sought to examine the utility of a ‘minimal panel’ for two reasons: “We combined the individual physiological observations that anticipated COVID-19 mortality trends to see if a ‘minimum panel’ could be useful for collection instead of the full NEWS panel. This makes our findings relevant to care home settings where NEWS is not calculated, or where all component physiological measures are not performed. Measurement of fewer components is expected to reduce contact time between carers and residents and decrease spread of COVID-19 infection”
- There does not seem to be a	Evidence on NEWS in care homes is limited. We sought

compelling reason to use 20th/80th centile scores. Why not simply follow the recommendations of the Royal College of Physicians? For example, they recommend “If the patient scores 3 in a component, regardless of the overall result, the patient is likely to require a higher level of care”.	to examine the use of NEWS as a population level surveillance tool - the RCP guidelines apply to measurements of individuals (usually in hospital settings): our analysis is on groups/clusters of groups aggregated to the regional level. P8 line5 “These cut points represent markers of increase in each parameter at a population level and are not measures of clinical concern at the individual resident level, which are defined elsewhere.⁴”
- When a patient has an above-baseline NEWS score it implies that the patient’s health has already deteriorated. And given that there is 2-week gestation period for COVID-19, it follows that there will be a spike in cases two weeks hence. What additional insights are generated beyond this are generated in the analysis? The analysis may foretell the coming increase in cases but may not offer any preventive solutions as these patients are likely infected already. Would it not be more beneficial, from a preventive perspective, to forecast above-baseline NEWS scores so that appropriate actions can be taken to prevent the spread of the virus?	We agree with the reviewer that our approach has limitations – many of which arise from the lack of individual level outcome information (a limitation we highlight in our discussion). It was beyond the aims (and scope) of this study to forecast NEWS due to limitations of our data, but our findings highlight that the collection of more data may assist in the future with forecasting. In light of the lack of widely available testing at the time of our analysis, we sought explore the use of NEWS as a surveillance tool. At time of writing COVID-19 is still causing excess deaths in UK care homes, despite a more developed testing system. https://www.theguardian.com/world/2021/jan/19/covid-related-deaths-in-care-homes-in-england-jump Examples of preventative measures that could be taken in response to patterns of elevated NEWS measurements could include more a more targeted/focussed testing regimen and a ‘stepping up’ of distancing or quarantining measures to reduce spread.
REVIEWER 2	
This is a very interesting and obviously timely paper in which the authors examine if there is any correlation between rise in NEWS	Thank you very much for your review, and suggestions – please see below for our responses to each of the points you raise.

measures and mortality in nursing homes in UK. I have the following suggestions to improve the paper:	
For the readers outside of the UK who are not familiar with the NEWS system, it would be very helpful if the authors include a section and explain the system, its history and maybe some areas that it has been proven effective before. Running this system smoothly requires significant effort in data collection and validation and I am very curious to know more about how the system is administrated and how the data is collected, verified and published.	We agree that international readers would benefit from more information about the history and use of NEWS. We have expanded on this in our introduction and added two references to the Royal College of Physician reports on NEWS. P5 – line 8 “This tool (originally developed by the Royal College of Physicians in 2012,³ and updated in 2017)⁴ is used in hospitals in the UK to identify patients at risk of acute deterioration and improve patient safety. NEWS requires the measurement of six simple physiological parameters: temperature, pulse, systolic blood pressure, respiratory rate and oxygen saturation, and level of consciousness or new confusion. The score generated should trigger a pre-specified response, ranging from repeating the score within a specific timeframe, to seeking urgent medical attention.⁵” We have also added a sentence and supporting reference to our methods to explain in greater detail how the measurements were recorded at scale. P7 – line 18 “NEWS recordings were made by care home staff using a specific, commercially available package designed to reduce measurement error. The commercial provider supplies equipment to measure vital signs and record physiological observations, automatically generate NEWS, and upload the data to their cloud storage servers. The system has previously been described elsewhere.¹¹”
I was also expecting to see more robust time series analysis in which the association between the two series (NEWS measures and nursing home mortality) is statistically examined, but the authors just present the graph, with no statistical analysis of the association between the two. This is easily doable in common statistical software, for example, the authors can build timeseries models	Thank you for highlighting this: we have clarified that we did use statistical modelling to estimate the cross correlation between the results and have highlighted the methods we used in the abstract and in the main body of the text. We have described correlations of interest in the results section, and supplemental figure 1 comprises 10 panels for cross-correlation function plots for NEWS (and component values) versus all cause deaths in care homes

very conveniently using TSFS tool in SAS and examine how the two series are correlated and if such correlation is statistically significant. Of course there are more involved methods such as VARMAX which requires more efforts and probably additional data which authors can explore.	in matched geographical areas. ABSTRACT P2 line 11 “Cross correlation comparison of time series with Office for National Statistics (ONS) weekly reported registered deaths of care home residents where COVID-19 was the underlying cause of death, and all other deaths (excluding COVID-19) up to 10/05/2020” P2 line 17 “We observed a rise in the proportion of above-baseline NEWS beginning March 16th 2020, followed two weeks later by an increase in registered deaths (cross correlation of $r=0.82$, $p \leq 0.05$ for a two week lag)” MAIN ARTICLE (Methods section) P8 line 20 “We calculated the cross-correlation between the two time series (above-baseline NEWS and component scores, and daily registered all cause deaths including COVID-19) for time lags between zero and seven weeks”
--	--

VERSION 2 – REVIEW

REVIEWER	Ram Gopal University of Warwick, Warwick Business School
REVIEW RETURNED	06-Feb-2021

GENERAL COMMENTS	The authors have adequately addressed my concerns and I am happy to recommend publication of the paper.
---

REVIEWER	Niam Yaraghi University of Miami, Business Technology
REVIEW RETURNED	12-Feb-2021

GENERAL COMMENTS	In the previous review, I had suggested the authors to do two things:  1- Provide an overview of the NEWS system 2- Conduct more rigorous time series analysis to justify their results in more robust way. The authors have neglected both of these recommendations. As a result, the paper lacks enough background and explanation of the setting it examines, neither does it succeed in providing a reliable and valid statistical analysis. The authors have also failed to include a response to reviewers, I suggest that they include a detailed response letter for their next revisions
--

VERSION 2 – AUTHOR RESPONSE

Reviewer: 1 Dr. Ram Gopal, University of Warwick Comments to the Author: The authors have adequately addressed my concerns and I am happy to recommend publication of the paper.	Thank you very much for your time and considering our revised manuscript. We're very pleased to see you have recommended publication of the paper
Reviewer: 2 Dr. Niam Yaraghi, University of Miami Comments to the Author: In the previous review, I had suggested the authors to do two things:	Thank you for considering our manuscript and providing thoughtful suggestions as to how it could be improved at the last round of review.
1- Provide an overview of the NEWS system	In our last submission, we agreed that international readers would benefit from more information about the history and use of NEWS. We expanded our introduction and added two references to the Royal College of Physician reports on NEWS. P5 – line 8 “This tool (originally developed by the Royal College of Physicians in 2012,³ and updated in 2017)⁴ is used in hospitals in the UK to identify patients at risk of acute deterioration and improve patient safety. NEWS requires the measurement of six simple physiological parameters: temperature, pulse, systolic blood pressure, respiratory rate and oxygen saturation, and level of consciousness or new confusion. The score generated should trigger a pre-specified response, ranging from repeating the score within a specific timeframe, to seeking urgent medical attention.⁵” We also added a sentence and supporting reference to our methods to explain in greater detail how the measurements were recorded at scale. P7 – line 18 “NEWS recordings were made

	by care home staff using a specific, commercially available package designed to reduce measurement error. The commercial provider supplies equipment to measure vital signs and record physiological observations, automatically generate NEWS, and upload the data to their cloud storage servers. The system has previously been described elsewhere.¹¹ We feel that the background and accompanying references now provide readers with sufficient detail to understand NEWS, and replicate our findings.
2- Conduct more rigorous time series analysis to justify their results in more robust way.	At the last revision, we clarified that we did use statistical modelling to estimate the cross correlation between the results, and clarified this in the methods, results and abstract. Cross-correlation is an appropriate and robust method of measuring the strength of an association between two time-series. ABSTRACT P2 line 11 “Cross correlation comparison of time series with Office for National Statistics (ONS) weekly reported registered deaths of care home residents where COVID-19 was the underlying cause of death, and all other deaths (excluding COVID-19) up to 10/05/2020” P2 line 17 “We observed a rise in the proportion of above-baseline NEWS beginning March 16th 2020, followed two weeks later by an increase in registered deaths (cross correlation of $r=0.82$, $p \leq 0.05$ for a two week lag)”

	MAIN ARTICLE (Methods section) P8 line 20 “We calculated the cross-correlation between the two time series (above-baseline NEWS and component scores, and daily registered all cause deaths including COVID-19) for time lags between zero and seven weeks” P8 line29 “used the ccf function in the stats R package to calculate cross-correlations” MAIN ARTICLE (Results section) P11 line 8 “The highest correlation was observed for a two-week lag (r=0.82, p< 0.05, supplemental figure 1)” SEE ALSO SUPPLEMENTAL INFORMATION FIGURE 1 FOR CROSS CORRELATION PLOTS FOR ALL VARIABLES OF INTEREST
The authors have neglected both of these recommendations. As a result, the paper lacks enough background and explanation of the setting it examines, neither does it succeed in providing a reliable and valid statistical analysis.	We were grateful for your thoughtful review and feel that we did attempt to address your concerns. The paper is now better for these revisions - thank you for your suggestions.
The authors have also failed to include a response The authors have also failed to include a response to reviewers, I suggest that they include a detailed response letter for their next revisions	We uploaded a document similar to this one alongside our response letter. In discussion with the editor, it seems as though you may not have been able to access this. We have uploaded the previous revision sheet to accompany this round of revisions too.